# Defects in Glutathione System in an Animal Model of Amyotrophic Lateral Sclerosis

**DOI:** 10.3390/antiox12051014

**Published:** 2023-04-27

**Authors:** Franziska T. Wunsch, Nils Metzler-Nolte, Carsten Theiss, Veronika Matschke

**Affiliations:** 1Department of Cytology, Institute of Anatomy, Ruhr-University Bochum, D-44801 Bochum, Germany; franziska.wunsch@rub.de (F.T.W.); carsten.theiss@rub.de (C.T.); 2International Graduate School of Neuroscience (IGSN), Ruhr-University Bochum, D-44801 Bochum, Germany; 3Inorganic Chemistry I—Bioinorganic Chemistry, Faculty of Chemistry and Biochemistry, Ruhr-University Bochum, D-44801 Bochum, Germany; nils.metzler-nolte@rub.de

**Keywords:** ALS, oxidative stress, wobbler mice, liver, hippocampus, cerebellum, glutamate cysteine ligase (GCL), glutathione synthetase (GSS), multidrug resistance protein (MRP), gamma-glutamyl transpeptidase (GGT)

## Abstract

Amyotrophic lateral sclerosis (ALS) is a progredient neurodegenerative disease characterized by a degeneration of the first and second motor neurons. Elevated levels of reactive oxygen species (ROS) and decreased levels of glutathione, which are important defense mechanisms against ROS, have been reported in the central nervous system (CNS) of ALS patients and animal models. The aim of this study was to determine the cause of decreased glutathione levels in the CNS of the ALS model wobbler mouse. We analyzed changes in glutathione metabolism in the spinal cord, hippocampus, cerebellum, liver, and blood samples of the ALS model, wobbler mouse, using qPCR, Western Blot, HPLC, and fluorometric assays. Here, we show for the first time a decreased expression of enzymes involved in glutathione synthesis in the cervical spinal cord of wobbler mice. We provide evidence for a deficient glutathione metabolism, which is not restricted to the nervous system, but can be seen in various tissues of the wobbler mouse. This deficient system is most likely the reason for an inefficient antioxidative system and, thus, for elevated ROS levels.

## 1. Introduction

Amyotrophic lateral sclerosis (ALS) is a progressive neurodegenerative disease characterized by the rapid degeneration of motor neurons in the motor cortex, brain stem, and spinal cord. To date, the pathogenesis and pathomechanisms of ALS have not been fully elucidated, with the majority of patients exhibiting signs of oxidative stress [1,2,3,4,5]. Similar to other neurodegenerative diseases, ALS is thought to be caused by a combination of genetic and environmental factors as well as age-related dysfunction [6]. The understanding of these genetic and pathophysiological mechanisms of ALS has led to the identification of promising therapeutic targets. However, over the past half-century, more than 50 randomized controlled trials (RCT) for proposed disease-modifying drugs have been negative [7]. Thus, no causative therapy exists so far [8].

Genetic studies of ALS have led to the development of numerous mouse models, which are highly relevant for understanding the pathophysiology of the disease and for the development of therapeutic targets. Among others, the wobbler mouse (WR) that spontaneously emerged in a C57BL/Fa strain is used as an ALS animal model [9]. These mice exhibit a spontaneous autosomal-recessive point mutation in the *vacuolar–vesicular protein sorting 54 (VPS54)* gene, which was soon linked to the degeneration of upper and lower motor neurons [10,11]. The point mutation results in a destabilization of the Golgi-associated retrograde protein (GARP) complex, which leads to an impairment of the retrograde vesicle transport and a missorting of endosomal and Golgi-associated proteins [12]. Golgi dysfunctions in wobbler mice have been related to various pathological abnormalities in motor neurons [13]. Recently, mutations in the *VPS54* gene have been discovered in ALS patients within Project MinE (http://databrowser.projectmine.com/, accessed on 24 April 2023) [14]. However, the exact role of Golgi dysfunction in the development of ALS remains to be investigated [12]. In homozygous wobbler mice, motor neuron death develops over time, and these mice mimic ALS symptoms, as seen in ALS patients [13]. The course of disease in wobbler mice can be divided into three phases [15]. In the pre-symptomatic phase, starting postnatal (p0), no clinical symptoms are present. Clinical symptoms arise during the evolutionary phase from p20 until p40. Here, wobbler mice progressively develop an unsteady gait, muscle weakness, and characteristic head tremors. In addition to this, wobbler mice remain smaller than their healthy littermates [16]. In the following symptomatic phase or distinct clinical stage, starting from p40, all symptoms fully develop and stagnate. The mice ultimately die from insufficiency of the respiratory musculature [15].

Elevated levels of reactive oxygen species (ROS) have been reported in the spinal cord tissue of wobbler mice [17,18] as well as in different cells derived from ALS patients [3,4,19]. The consequences of oxidative stress, such as damage to DNA, proteins, and lipid membranes, can be found in ALS patients [1,5,20,21,22,23] as well as in different ALS mouse models [18,24,25,26,27].

For maintaining the cells’ redox balance, an antioxidant detoxification system is vitally important [28]. This includes endogenous antioxidant enzymes and co-enzymes, such as glutathione and ubiquinone, as well as exogenous dietary antioxidant molecules, such as ascorbic acid and tocopherols. Glutathione is the most abundant antioxidant in the cell [28]. Reduced levels of glutathione are not only present in apoptosis but also precede neurodegeneration [29]. The bulk of plasma glutathione originates from the liver as hepatocytes export glutathione into plasma for inter-organ glutathione homeostasis [30]. Plasma glutathione has been reported to influence glutathione uptake at the blood–brain barrier [31,32,33]. Unlike astrocytes, neurons do not have an uptake mechanism for glutathione and, therefore, rely on their own synthesis to maintain adequate concentrations. Astrocytes play an essential role in providing neurons with substrates for glutathione synthesis [34]. In the central nervous system (CNS), glutathione is cleaved into its constituent amino acid in the extracellular space by enzymes located in the plasma membrane of different cell types of the CNS [35]. The generated cysteine and glycine can then be imported into other cells, such as neurons and oligodendrocytes, via different amino acid transporters [36]. Notably, cysteine has been reported to be the rate-limiting precursor for glutathione synthesis in neurons [37,38]. In the first step of glutathione synthesis, the enzyme γ-Glutamyl-cysteinyl-ligase (GCL), which consists of a catalytic (GCLC) and a modifier subunit (GCLM), creates γ-glutamyl-cysteine from the amino acids glutamate and cysteine under consumption of adenosine triphosphate (ATP). This first step of GSH synthesis is considered to be rate-limiting [39]. The next step is catalyzed by the glutathione synthetase (GSS). It adds glycine to the precursor γ-glutamyl-cysteine under the consumption of ATP, and glutathione is formed.

Accumulating evidence suggests that aberrant glutathione homeostasis is linked to the development and progression of ALS, as recently reviewed by Kim [40]. This includes low levels of glutathione in erythrocytes of ALS patients [41] and, importantly, the detection of reduced levels of glutathione by 31% in the precentral gyrus of ALS patients compared to healthy volunteers, using a magnet resonance (MR) spectroscopic technique [42]. Another MR spectroscopic imaging study revealed that glutathione levels in the motor cortex and corticospinal tract of ALS patients inversely correlate with time from diagnosis to imaging [43]. On the molecular level, reduced levels of glutathione and a decreased ratio of reduced glutathione (GSH)/oxidized glutathione (GSSG) in whole blood and the lumbar spinal cord have been reported in G93A-SOD1 transgenic mice [44,45] as well as in cervical spinal cord of wobbler mice [18].

In this study, we aim to determine the cause of glutathione depletion in the cervical spinal cord of wobbler mice. Therefore, we compared factors of glutathione synthesis and ROS levels in different organ systems (liver, blood) and different CNS areas (spinal cord, hippocampus, cerebellum) to clarify whether defects in glutathione homeostasis in wobbler mice are restricted to motor areas of the CNS.

## 2. Materials and Methods

### 2.1. Animals

All animal experiments were performed in accordance with EU guidelines 2010/63/EU concerning the protection of animals used for scientific purposes. Animal experiments were conducted according to German animal welfare regulations and approved by the local authorities (registration number Az. 84-02.04.2017.A085). Breeding, handling, and genotyping were carried out as described previously [16]. Mice always had free access to food and water and were kept at a 12-h night-and-day cycle. For each experiment, three to six homozygous wild-type (wt/wt) and wobbler (wr/wr) mice in the distinct clinical phase (p40) were used. Both genders were used.

### 2.2. Sample Collection

For qPCR, Western Blot, glutathione, and ROS analysis, mice were sacrificed by decapitation. The cervical part of the spinal cord, liver, hippocampus, and cerebellum were isolated, snap-frozen in liquid nitrogen, and stored at −80 °C. For the collection of whole blood, erythrocyte, and plasma samples, mice were deeply anesthetized with a combination of Ketamin (100 mg/kg) and Xylazin (10 mg/kg). After thoracic dissection, blood was collected from the left ventricle in a K3EDTA laminated vessel (#20.1341, Sarstedt, Nümbrecht, Germany). For erythrocyte and plasma collection, samples were centrifuged at 2000× *g* at 4 °C for 15 min. Afterward, the plasma and erythrocyte fraction were transferred to a clean tube. Blood samples were stored at −80 °C.

### 2.3. RNA-Isolation, Reverse Transcription, and Quantitative PCR

Total ribonucleic acid (RNA) was isolated from 15–20 mg of frozen tissue using the ReliaPrep™ RNA Tissue Miniprep System (#Z6111, Promega, Madison, WI, USA) following the manufacturer’s protocol. Afterward, 500 ng of purified RNA was transcribed into complementary desoxyribonucleic acid (cDNA) using the GoScript™ Reverse Transcription Mix, Oligo(dT) (#A2791, Promega, Madison, WI, USA) according to the manufacturer’s instructions. The quantitative real-time polymerase chain reaction (qPCR) was conducted with 400 ng of cDNA. qPCR was performed using the GoTaq^®^ qPCR Master Mix (#A6001, Promega) on a CFX96 Real-Time PCR Detection System (Bio-Rad, Hercules, CA, USA). Specific primers used are shown in Table 1. Gene expression was investigated in triplicates and normalized to *Glyceraldehyde–phosphate dehydrogenase* (*GAPDH)*. The analysis was performed with tissue samples collected from at least four wild-type (WT) and four wobbler mice (WR). The obtained cycle threshold (Ct)-values were analyzed using the 2^−ΔΔCt^—method [46].

### 2.4. SDS-PAGE and Western Blot

For protein isolation, 15–20 mg of frozen tissue was used. Tissue samples were subjected to Radioimmunoprecipitation Assay (RIPA) buffer (#9806, Cell Signaling Technology, Danvers, MA, USA), supplemented with protease inhibitor (#11873580001, Roche, Mannheim, Germany) and homogenized with a clearance pestle and a 27G syringe. Samples were kept on ice the whole time. Next, samples were centrifuged at 12,700× *g* for 15 min at 4 °C, and the supernatant was transferred to a clean tube. Protein concentration was determined by Pierce^TM^ BCA Protein Assay Kit (#23225, Thermo Fisher Scientific, Waltham, MA, USA) following the manufacturer’s protocol. A 4× Laemmli sample buffer (#161-0747, BioRad, Hercules, CA, USA) was added to the samples. For detection of GCLC, GSS, and gamma-glutamyl transferase 1 (GGT1), 50 µg and for multidrug resistance protein 4 (MRP4), 100 µg of total protein per lane were separated by sodium dodecyl–sulfate–polyacrylamide gel electrophoresis (SDS-PAGE). Next, proteins were transferred to membranes (Table 2) via Trans-Blot Turbo Transfer System (BioRad, Hercules, CA, USA). All membranes were blocked for 2 h at room temperature. Next, the primary antibodies (Table 2) were incubated at 4 °C overnight. For GCLC and MRP4, actin was chosen as a housekeeper. For protein detection of GSS and GGT1, Calnexin was chosen as a housekeeper. The next day, membranes were washed with Tris-Buffered Saline with 0.1%Triton^TM^ X-100 (TBS-T) and incubated with the appropriate horseradish–peroxidase-conjugated secondary antibody (Table 2) for two hours at room temperature. After four washes with TBS-T and two washes with Tris-Buffered Saline (TBS), protein bands were visualized with Western Blotting Luminol Reagent (#sc-2048, Santa Cruz, Dallas, TX, USA) in an imaging system (ChemiDoc XRS+, Bio-Rad, Hercules, CA, USA). For semiquantitative analyses, the band intensities of the proteins of interest were normalized to the loading control. All values were then normalized to the mean value of the normalized WT signals and plotted as a percentage in a bar chart. Western Blot analysis was performed with tissue samples obtained from at least four wild-type (WT) and four wobbler mice (WR).

### 2.5. HPLC

Total glutathione (tGSH) concentrations in whole blood were assessed by high-performance liquid chromatography (HPLC) separation and fluorescent detection using a glutathione HPLC kit (#KC 1800, Immundiagnostik, Bensheim, Germany), according to manufacturer’s instructions. In brief, ethylenediaminetetraacetic acid (EDTA) whole blood samples were treated with an internal standard and a reducing agent. Samples were derivatized, precipitated, and treated with a reaction buffer. Then, 20 µL was injected into an HPLC system. The separation via HPLC followed an isocratic method at 30 °C and a flow rate of 1 mL/min using a reversed-phase column (NUCLEODUR 100-3 C18 ec, 3 µm, 125 × 4 mm, #760051.40, Macherey–Nagel, Düren, Germany). Chromatograms were recorded by a fluorescence detector using an excitation wavelength of 385 nm and an emission wavelength of 515 nm. Quantification was performed with an EDTA-blood calibrator included in the kit. Low-concentration and high-concentration control samples also included in the kit, were measured as reference material. Samples, controls, and calibrators were measured twice. Reference values: tGSH 763–1191 µmol/L. Intra-assay coefficient of variation (CV): tGSH 3.9%. Inter-assay CV: tGSH 4.2%. tGSH concentrations were calculated via the peak height by the internal standard method. HPLC analysis was performed with whole blood drawn from six wild-type and six wobbler mice.

### 2.6. Glutathione Assay

For measurement of glutathione in tissue, erythrocyte, and plasma samples, a GSH/GSSG Ratio Detection Assay Kit (#ab205811, Abcam, Cambridge, UK) was used, according to the manufacturer’s instructions. In brief, 20 mg of tissue, 3 µL of erythrocytes, and 6 µL of plasma were diluted 1:20, 1:100, and 1:35, respectively, in ice-cold phosphate-buffered saline (PBS) containing 0.5% Nonidet P40 (PBS/0.5% NP40). Tissue samples were homogenized and centrifuged at 13,000× *g* at 4 °C for 15 min. Fifty µL of each sample were plated in duplicates in a 96-well plate suitable for fluorescence measurement. For measurement of reduced GSH only, 50 µL of GSH Assay mixture (GAM) was added to each well. For measurement of tGSH, 50 µL of total glutathione Assay mixture (tGAM) was added to each well. With the aid of tGAM, which contains nicotinamide adenine dinucleotide phosphate (NADPH) and Glutathione reductase, GSSG is converted to tGSH in an enzymatic reaction with a ratio of 1 mole GSSG to 2 moles of GSH. After an incubation of 25 min at room temperature, protected from light, fluorescence was read with a Cytation^TM^ 5 imaging reader (BioTek Instruments, Winooski, VT, USA) at 490 nm excitation and 520 nm emission with a reading time of 100 ms. The concentration of reduced GSH can be calculated from a GSH standard curve. The concentration of oxidized GSSG is calculated as follows:(1)$$\left[GSSG\right]=\frac{\left(tGSH-GSH\right)}{2}$$

The glutathione assay was performed with tissue, erythrocyte, and plasma samples obtained from at least four wild-type and four wobbler mice.

### 2.7. ROS Assay

For the detection of ROS in tissue samples, OxiSelect^TM^ ROS/RNS In Vitro Assay Kit (#STA-347, Cell Biolabs, San Diego, CA, USA) was used. In brief, 10 mg/mL of tissue was homogenized in ice-cold 1xPBS and centrifuged at 10,000× *g* for 5 min at 4 °C right after extraction. Homogenates were stored at −80 °C until use. The assay was performed according to the manufacturer’s instructions. In brief, 50 µL of each sample was plated in triplicates in a 96-well plate suitable for fluorescence measurement. Then, 50 µL of catalyst was added to each well and incubated for 5 min at room temperature. Afterward, 100 µL of 2′-7′dichlorofluorescin diacetate (DCFH) solution was added to each well. Fluorescence was measured after an incubation of 20 min at room temperature, protected from light, using a Cytation^TM^ 5 imaging reader (BioTek Instruments, Winooski, VT, USA) at 480 nm excitation and 530 nm emission with a reading time of 100 ms. ROS assay was conducted with tissue samples collected from four wild-type and four wobbler mice.

### 2.8. Statistical Analysis

Prism 6 (GraphPad Inc., La Jolla, CA, USA) was used for statistical analysis. Data represent mean values of at least three independent experiments ± standard error of the mean (SEM). Data were tested for significance using Student‘s *t*-test, and results with *p*-values < 0.05 were considered statistically significant.

## 3. Results

There is accumulating evidence pointing toward the involvement of reactive oxygen species in the neurodegenerative processes of ALS [1,3,4,5,19,20,21,22,23]. As glutathione is an important defense system against ROS and previous studies have reported reduced glutathione levels in ALS [41,42,43], we aimed to analyze glutathione metabolism in symptomatic wobbler mice in detail.

### 3.1. Physiological Glutathione Synthesis in Liver of Wobbler Mice

First, we aimed to investigate whether deficits already exist in the global glutathione production of the main organ for GSH synthesis, the liver. Therefore, we investigated mRNA and protein expression of enzymes involved in glutathione synthesis in the liver during the distinct clinical stage (p40) via qPCR and Western blot.

Gene expression of the catalytic (*Gclc*) and the modifier (*Gclm*) subunit of GCL was significantly upregulated in wobbler mice, while mRNA expression of *Gss* was not altered (Figure 1a). However, on the protein level, GCLC showed no alteration in wobbler mice compared to wild-type mice (Figure 1b), presumably resulting in an unaltered production of GSH precursor γ-glutamyl-cysteine. Interestingly, protein expression of GSS was significantly increased (Figure 1b). Protein expression of the cleaving enzyme GGT1 was also unaltered in the liver tissue of p40 wobbler mice compared to wild-type littermates (Figure 1b). Measurements of total GSH (tGSH), reduced form of glutathione (GSH), and oxidized form of glutathione (GSSG) levels showed no significant differences in liver tissue of p40 wobbler mice compared to wild-type mice (Figure 1c,d). The observed unaltered levels of tGSH and the GSH/GSSG ratio indicate that the liver of wobbler mice is not exposed to oxidative stress. This was confirmed by the following measurement of the levels of reactive oxygen species within the two genotypes. ROS levels in liver tissue of p40 wobbler and wild-type mice showed no significant alteration (Figure 1e).

### 3.2. Decreased Levels of Glutathione in Plasma and Erythrocytes of Wobbler Mice

Based on our findings in the liver tissue of wobbler mice, we hypothesized that physiological amounts of glutathione are released into the plasma by hepatocytes. For this reason, we analyzed glutathione amounts in whole blood, plasma, and erythrocytes of wobbler mice during the distinct clinical stage (p40). In whole blood, tGSH levels were analyzed with the aid of HPLC separation and fluorometric detection. tGSH levels in whole blood of p40 wobbler mice showed no significant alterations compared to age-matched wild-type controls (Figure 2a). A fluorometric glutathione assay was used for the measurement of tGSH, GSH, and GSSG levels in plasma and erythrocytes. In plasma, wobbler mice displayed significantly lower levels of tGSH as well as a significantly decreased ratio of GSH/GSSG compared to p40 wild-type mice (Figure 2b,c). tGSH levels were unaltered in the erythrocytes of wobbler mice compared to wild-type mice at the age of p40 (Figure 2d). In line with the findings in plasma, the ratio of GSH/GSSG in erythrocytes of p40 wobbler mice was significantly decreased compared to age-matched controls (Figure 2e).

### 3.3. Altered Expression of Enzymes Involved in Transport of Glutathione and Its Components in the Cervical Spinal Cord

In the next step we aimed to find out how the detected deficiency of plasma glutathione in wobbler mice is linked to the glutathione deficit in the cervical spinal cord reported by our group before [18]. First, we investigated gene expression of enzymes involved in transport of glutathione and its components in the cervical spinal cord of wobbler mice in the distinct clinical stage (p40) via qPCR (Figure 3a). mRNA expression of Mrp4, which has been assumed to transport glutathione across the blood–brain barrier [31,47], was not significantly altered (Figure 3a). However, multidrug resistance protein 1 (Mrp1), which is involved in glutathione export from astrocytes [48], showed a significantly elevated gene expression (Figure 3a). Unfortunately, we were unable to obtain a reliable Western Blot for MRP4 protein expression in spinal cord.

Besides the transport of glutathione itself, the import of amino acids is crucial for glutathione synthesis in order to achieve adequate glutathione concentrations in cells of the CNS. The cystine/glutamate exchanger system Xc- consists of a light chain xCT, encoded by *Solute carrier (Slc) 7a11 (Slc7a11)*, and a heavy chain 4F2hc, encoded by *Slc3a2* [49]. mRNA levels of *Slc7a11* were significantly elevated in wobbler mice, while mRNA levels of *Slc3a2* were not altered (Figure 3a). As glutathione synthesis takes place in the cytosol, GSH is imported into mitochondria via 2-oxoglutarate/malate carrier protein, which is encoded by *Slc25a11* [50]. mRNA levels of *Slc25a11* in the cervical spinal cord of wobbler mice showed no alterations compared to wild-type mice (Figure 3a). Unlike astrocytes, neurons do not contain an uptake mechanism for glutathione and, therefore, rely on their own synthesis [34]. We analyzed mRNA expression of excitatory amino acid transporter 3 *(Eaat3)*, the major transporter for the import of cysteine into neurons [36]. Remarkably, we found a significant downregulation of *Eaat3* mRNA in the cervical spinal cord of wobbler mice (Figure 3a).

### 3.4. Upregulation of Enzymes Involved in Glutathione Cleavage in the Cervical Spinal Cord of Wobbler Mice

GSH released by astrocytes is cleaved to its constituent amino acids, which are later imported into neurons. For this reason, we next analyzed the gene expression of enzymes involved in glutathione cleavage in the cervical spinal cord of wobbler mice via qPCR (Figure 3b). First, GSH is hydrolyzed by the ectoenzyme γ-glutamyl-transferase (GGT) in the extracellular space [38]. Interestingly, the gene expression of *Ggt6* was significantly upregulated, while the gene expression of *Ggt1* and *Ggt7* was unaltered (Figure 3b). However, on the protein level, we found a significantly upregulated expression of GGT1 in the cervical spinal cord of p40 wobbler mice (Figure 3d). In the next step, the ectoenzyme aminopeptidase N (Anpep) further cleaves the resulting cysteinyl–glycine [35]. Remarkably, gene expression of *Anpep* was significantly upregulated (Figure 3b).

### 3.5. Decreased Levels of GCLC in the Cervical Spinal Cord of Wobbler Mice

The results described above demonstrate an upregulation of up to 40% of the investigated genes of enzymes involved in glutathione metabolism in the cervical spinal cord of p40 wobbler mice. Therefore, we next analyzed the gene expression of enzymes involved in glutathione synthesis in the cervical spinal cord of wobbler mice in the distinct clinical stage (p40) via qPCR. mRNA expression levels of *Glutaminase1 (Gls1)* and *Gclm* were unaltered, while mRNA expression levels of *Gclc* and *Gss* were significantly decreased in wobbler mice compared to wild-type (Figure 3c). To confirm whether this also applies to the protein level, a Western Blot analysis was performed. Here, protein expression of GCLC was significantly decreased in the cervical spinal cord of p40 wobbler mice (Figure 3d). However, Western Blot analysis of GSS displayed significantly increased protein expression in the cervical spinal cord of p40 wobbler mice compared to wild-type mice (Figure 3d).

### 3.6. Physiological Glutathione Metabolism in Hippocampus and Cerebellum of Wobbler Mice

To verify whether the observed changes in glutathione metabolism are specific to the spinal cord or if they also occur in other areas of the CNS of the wobbler mice, we performed the same analysis on hippocampal and cerebellar tissue from p40 wobbler and wild-type mice.

In the hippocampus, qPCR revealed an upregulation of all analyzed genes involved in the transport of glutathione and its components in wobbler mice compared to wild-type mice (Figure 4a), whereas Western Blot analysis of MRP4 indicated an unaltered protein expression (Figure 4b). mRNA expression levels of genes involved in glutathione cleavage in the hippocampus were not significantly altered or even decreased (Figure 4a). Protein expression of the cleaving enzyme GGT1 was unchanged in the hippocampus of p40 wobbler mice compared to wild-type mice (Figure 4b). For enzymes involved in glutathione synthesis in the hippocampus, qPCR showed significantly increased mRNA expression levels of *Gclc,* besides unaltered mRNA expression levels of *Gclm* and *Gss* (Figure 4a). However, on the protein level, Western Blot analysis revealed an unaltered expression of GCLC and a significantly upregulated expression of GSS (Figure 4b). Measurements of glutathione level in hippocampal tissue of p40 wobbler mice confirmed the assumption of unaltered glutathione synthesis in the hippocampus, as no significant alterations compared to wild-type mice were obtained (Figure 4c,d). Nevertheless, ROS assays of hippocampal tissue detected a modest but significant increase of reactive oxygen species in the hippocampus of wobbler mice compared to wild-type mice (Figure 4e).

In the cerebellum of p40 wobbler mice, qPCR showed a significant upregulation of all analyzed genes involved in the transport of glutathione and its components (Figure 5a). For enzymes involved in glutathione cleavage, only *Anpep* and *Ggt6* showed a significantly upregulated gene expression (Figure 5a). In contrast to mRNA expression, Western Blot analysis revealed a significantly decreased protein expression of MRP4 and a significantly increased protein expression of the cleaving enzyme GGT1 in the cerebellum of p40 WR mice compared to wild-type mice (Figure 5b). We also analyzed enzymes involved in glutathione synthesis in the cerebellum of p40 wobbler and wild-type mice. Here, mRNA levels of *Gclc* were significantly increased (Figure 5a). However, protein expression of GCLC was not significantly altered, while GSS was significantly increased (Figure 5b). The hypothesis of a sufficient metabolism and synthesis of glutathione in the cerebellum of p40 wobbler mice was confirmed by glutathione and ROS assays, which showed neither alterations in glutathione (Figure 5c,d) nor in ROS (Figure 5e) levels in cerebellum of p40 wobbler mice compared to wild-type mice.

## 4. Discussion

Increasing evidence points to a disturbed glutathione redox balance in regard to the development and progression of ALS [40]. As the glutathione system is an important antioxidant system within the cells, any change in homeostasis can have severe consequences for the well-being of the cells [51]. Since it is meanwhile known that signs of oxidative stress are present in both tissue and motor neurons of ALS animal models [17,18,24,25,26,27] as well as in postmortem tissue, plasma, urine, and cerebrospinal fluid of ALS patients [1,3,4,5,19,20,22,23,52], the intrinsic antioxidative systems seem to be functioning insufficiently. Furthermore, our previous study showed a reduced amount of total glutathione in the spinal cord of wobbler mice, lacking adequate protection against oxidative stress [18]. However, the precise mechanism leading to glutathione depletion remains to be determined. In our current study, we demonstrated that glutathione level is reduced in the spinal cord tissue of the ALS model wobbler mouse due to a decreased expression of the rate-limiting enzyme of the glutathione synthesis, GCLC, in addition to defective glutathione metabolism.

The liver has been reported to mainly determine the plasma glutathione concentration [30], which is positively correlated to the glutathione uptake across the blood–brain barrier [31,32,33]. Furthermore, ultrastructural changes in the liver of ALS patients have been reported before [53,54]. For this reason, the liver and plasma were examined for defects in glutathione content and synthesis. Our results show an unaltered or even increased protein expression of GGT1, GCLC, and GSS, respectively, leading to an unaltered glutathione cleavage and synthesis in the liver of symptomatic wobbler mice compared to wild-type mice. In accordance with the unaltered glutathione levels as well as unaltered ROS levels measured in the liver of symptomatic wobbler mice, we concluded that appropriate contents of glutathione are produced and hypothesize a physiological release of GSH by hepatocytes into the plasma. In whole blood and erythrocytes, we detected unchanged levels of total glutathione amount. Approximately 98% of glutathione in whole blood is localized inside erythrocytes as plasma glutathione is utilized for interorgan glutathione homeostasis [55,56]. Our results show a decreased ratio of GSH/GSSG in erythrocytes of symptomatic wobbler mice, indicating increased oxidation of GSH due to oxidative stress. As a result, GSH is decreased, which is in line with the finding of Babu et al., who reported decreased levels of GSH in erythrocytes of ALS patients [41]. Blasco et al. reported comparable results as they found an increased ratio of GSSG/GSH in the whole blood of ALS patients [52]. The unchanged total glutathione levels in erythrocytes of wobbler mice suggest a physiological glutathione synthesis in these cells. Interestingly, plasma glutathione levels were decreased in symptomatic wobbler mice compared to wild-type mice. Not only the total glutathione content in plasma was decreased, but also the ratio of GSH/GSSG, indicating a defective redox balance. Glutathione is sustainably exported from erythrocytes into plasma under physiological circumstances [56]. We assume a functioning export of glutathione from the liver into plasma as described above. This leaves us with two possible explanations for the decreased levels of total glutathione in the plasma of symptomatic wobbler mice: firstly, due to the disturbed redox balance in erythrocytes described above, their export of glutathione into plasma might be restricted [56]. Secondly, a considerable amount of plasma glutathione might be utilized to support glutathione homeostasis in neighboring cells and tissue [55]. It, thus, appears that changes in glutathione metabolism are not limited to the CNS in wobbler mice but also occur in other tissues. It is tempting to speculate that due to decreased levels of plasma glutathione, not enough glutathione is available for uptake across the blood–brain barrier [33]. Nevertheless, the impact of plasma glutathione on glutathione levels in the CNS remains controversial [57]. Interestingly, López-Blanch et al. reported that in isolated hepatocytes of two other ALS mouse models, SOD1G93A and FUS-R521C mice, GSH synthesis and efflux significantly increased while GSH concentration decreased in whole blood in both models at an advanced state of disease progression [58]. Considering the reported decreased levels of glutathione in the blood of ALS patients and different mouse models, including our results, alterations in blood glutathione appear to apply to more than one ALS genotype as a promising therapeutic approach in the future [41,52,58].

Glutathione transport across the blood–brain barrier is likely mediated by Multidrug Resistance Protein 4 (MRP4) [31,47]. Quantitative proteomics of MRP expression in brain capillary endothelial cells showed that only MRP4 is expressed at quantifiable levels at the blood–brain barrier [47,59]. In addition to glutathione, cysteine, and cystine uptake have been reported to influence glutathione contents in the central nervous system [34]. Cysteine, which is considered the rate-limiting amino acid for glutathione synthesis [60], can be imported into cells directly or in its oxidized form cystine via the cystine/glutamate exchanger system Xc- [61]. System Xc- is mainly expressed in glial cells [62] and is highly inducible by oxidative stress [63]. Astrocytes play an essential role in providing neurons with substrates for GSH synthesis [34]. GSH is released by astrocytes via MRP1 [48] and gap junction hemichannels [64]. In the extracellular space, glutathione is hydrolyzed by the ectoenzyme γ-glutamyl-transferase [38], located in the plasma membrane of non-neuronal cells [65]. In the next step, the ectoenzyme aminopeptidase N further cleaves the resulting cysteinyl–glycine [35]. The generated cysteine and glycine can then be taken up by other cells, such as neurons and oligodendrocytes, via different amino acid transporters.

In the current study, we provide evidence that the reduced total glutathione contents in the cervical spinal cord of symptomatic wobbler mice reported earlier by our group [18] are caused by a reduced glutathione synthesis due to decreased protein expression of the rate-limiting enzyme in glutathione synthesis GCLC [39]. We assume this leads to reduced production of GSH precursor γ-glutamyl-cysteine, resulting in a decreased GSH production, although protein expression of GSS is upregulated. Similar to Junghans et al. [18], we studied tissue samples of the cervical spinal cord of wobbler mice and, thus, cannot differentiate between the different cell types, which should be addressed in the future. However, we can assume a possible mechanism based on the specific expression of the different proteins involved in glutathione metabolism in the cervical spinal cord (Figure 6). We hypothesize that less GSH is released by astrocytes due to reduced synthesis. Presumably compensatory, *Mrp1* mediating GSH export from astrocytes [48] is upregulated, resulting in less GSH available for cleavage in the extracellular space. We suspect that, as a compensatory mechanism, the cleaving enzymes *Anpep* and GGT1 are upregulated in the cervical spinal cord. GGT plays an essential role in the astrocytic supply of components for glutathione synthesis to neurons [38,66]. Inhibition of GGT decreased intracellular glutathione contents by about 25% in dissociated neuronal cultures [66]. Similarly, in transient neuron-astroglia co-cultures, glutathione content in the neuronal compartment increased by 165% after the application of astrocytes, while inhibition of GGT completely prevented this effect [38]. Moreover, several studies have demonstrated that GGT mRNA and protein expression, as well as its activity, increase after exposure to oxidative stress [67,68,69]. Aminopeptidase N is involved in the degradation and modulation of several peptides, including glutathione precursor cysteinyl–glycine [35,70]. As GGT is more specifically involved in GSH degradation, we focused on its protein expression. However, our results are limited to the fact that we cannot discuss protein activity. Finally, decreased GSH synthesis and release by astroglial cells lead to a decreased supply of amino acids essential for glutathione synthesis in neurons. In accordance with our result of reduced mRNA levels of *Eaat3*, decreased protein expression of EAAT3 has been reported in the cervical spinal cord of symptomatic wobbler mice [71]. As EAAT3 has been described to be the major transporter for cysteine import in neurons [36], its downregulation aggravates the deficient cysteine supply to neurons [37]. In cultured neurons, cysteine uptake and intracellular GSH contents were reduced by inhibition of EAAT3 but not by inhibition of other cysteine transporters [72]. EAAT3 null mice display reduced glutathione contents, increased oxidant levels, and suffer from age-dependent neurodegeneration along with cognitive impairments [36]. These changes could be reversed by treating the mice with N-Acetylcysteine, a membrane-permeable cysteine precursor [36]. We hypothesize that overall, the decreased glutathione synthesis due to a decreased expression of the rate-limiting enzyme GCLC and the release by astrocytes, as well as a decreased cysteine uptake in neurons, lead to an impaired glutathione synthesis in the cervical spinal cord. Due to the inadequate glutathione contents, neurons’ and astrocytes’ defense against ROS is severely impaired. Consecutively, this leads to elevated levels of ROS in the cervical spinal cord [17] and particularly in motor neurons of wobbler mice [18].

Interestingly, Riluzol, the only drug approved by the European Medical Agency for the treatment of ALS with a small significant effect on survival, has been reported to augment EAAT3 expression and decrease ROS in C6 astroglial cells [73]. In neurons, glutamate and cysteine transport capacities are regulated by EAAT3 recycling [74,75]. EAAT3 recycling from the plasma membrane is mediated by clathrin-dependent endocytosis, followed by recycling through Rab11-positive vesicles, from which transporter molecules can be dynamically mobilized to the cell surface to mediate cysteine uptake [76,77]. Defects in Rab11-dependent trafficking of EAAT3 to the cell surface have been reported in primary cortical neurons in a knock-in mouse model of Huntington’s disease and result in impaired cysteine uptake and glutathione biosynthesis [78]. The wobbler mouse displays a point mutation affecting VPS54, a component of the Golgi-associated retrograde protein (GARP) complex, tethering endosome-derived vesicles to the trans-Golgi network [12]. Recently, Wilkinson discovered that the specific disruption of both *scattered* (*scat*), the fly VPS54 ortholog, and *Rab11* gene expression result in muscle atrophy and reduced climbing ability in Drosophila [79]. Thus, there might be a connection between the VPS54 mutation and disturbed EAAT3 recycling at the plasma membrane, aggravating an impaired cysteine supply and glutathione synthesis in neurons. However, the exact mechanism of how the VPS54 mutation results in the ALS seen in wobbler mice remains to be explored.

Based on the findings in the cervical spinal cord, we wondered whether deficits in glutathione homeostasis are limited to motoric areas of the CNS in wobbler mice. For this reason, we analyzed tissue samples of the cerebellum and hippocampus of p40 wobbler mice. Fifty percent of ALS patients suffer from additional extra motor manifestations, among them progressive cognitive abnormalities [6]. Fifteen percent of ALS patients meet the diagnostic criteria for frontotemporal dementia [80]. Repeat expansions in *chromosome 9 open reading frame 72* (*C9ORF72*) are currently the most important genetic cause of familial ALS and frontotemporal dementia, accounting for approximately 34.2% and 25.9% of the cases [81]. In accordance with this, hippocampal abnormalities have been identified in ALS patients [82] and wobbler mice [83]. Our results show unaltered glutathione levels besides an unaltered glutathione synthesis in the hippocampus of p40 wobbler mice compared to wild-type mice. Nevertheless, ROS levels were modestly elevated in the hippocampus of p40 WR mice. This gives rise to two possible interpretations: either the glutathione system is balanced in the hippocampus, and there is another cause for ROS production, or there is a glutathione deficit in some hippocampal cells, which is counterbalanced by other cells, leading to the measurement of unaltered levels of the bulk glutathione concentrations. As we measured the bulk of ROS and glutathione content in hippocampal tissue, we could not differentiate these levels in the different cell types and the extracellular space. For this reason, our measurements are likely to underestimate the degree of glutathione deficiency in neurons because a large share of brain glutathione is located in astrocytes [84]. Therefore, we cannot exclude a glutathione deficiency specifically in neurons nor in other cell types in the hippocampus of wobbler mice leading to oxidative stress. The upregulated gene expression of enzymes involved in glutathione transport in hippocampus might point towards this hypothesis. For this reason, future studies are needed to specifically differentiate glutathione and ROS levels in the different cell types.

Previous studies have revealed alterations in the cerebella of p40 wobbler mice [85,86]. Besides inflammatory processes, such as astrogliosis and microgliosis, no signs of neurodegeneration have been detected [86]. In addition to this, there is accumulating evidence of cerebellar involvement in ALS [87]. Focal cerebellar degeneration and cerebro–cerebellar connectivity alterations have been described in ALS patients in vivo via magnetic resonance imaging [88]. However, it remains unclear if cerebellar involvement is a primary or rather secondary compensatory mechanism as the motor cortex degenerates [89]. Our results demonstrate unaltered levels of ROS and glutathione in the cerebella of p40 wobbler mice compared to wild-type littermates. These results are in line with previous findings concerning the cerebellum of ALS mouse models. Aguirre et al. detected no oxidative DNA damage in the cerebellum of G93A SOD1 mice [27]. In symptomatic wobbler mice, Saberi et al. observed an inflammatory response but did not identify evidence of neuron cell death, while Klatt et al. identified a downregulation of proapoptotic factors linked to neuroprotection [85,86]. In addition to this, we report upregulated mRNA expression of enzymes involved in glutathione transport and cleavage, as well as upregulated protein levels for MRP4 and GGT1. We hypothesize this is an adaption due to the inflammatory processes described earlier [86]. This result is limited to the fact that for some genes, the verification remains to be determined on the protein level. Interestingly, these upregulated genes encode stress-inducible GSH metabolic enzymes whose induction is mediated by Nrf2 (nuclear factor erythroid 2-related factor 2) via ARE (antioxidant responsive element) [63,90,91]. The Nrf2 signaling pathway regulates the cellular response to stress and inflammation [92]. This might be the reason for the detected upregulation of genes and proteins encoding stress-inducible metabolic enzymes. Overall, we conclude that in the cerebellum of symptomatic wobbler mice, redox homeostasis is balanced.

Our data present decreased levels of GCLC in the cervical spinal cord but not in the cerebellum or hippocampus of wobbler mice. Key transcription factors involved in the regulation of Gclc gene expression are Nrf2 via ARE, NFκB (nuclear factor kappa B), and AP-1 (activator protein-1) [93]. Usually, oxidative stress leads to the induction of genes involved in antioxidative defense, including *Gclc* [93]. However, expression of *Gclc* and *Gclm* can become dysregulated if oxidative conditions persist, as reported in liver injuries [94]. Therefore, we hypothesize that a decreased mRNA and protein expression of GCLC in the cervical spinal cord is due to prolonged oxidative stress. Feng et al. reported that selective deficiency of *Gclc* in mouse CNS leads to neurodegeneration, specifically progressive motor neuron loss marked by mitochondrial dysfunction [95]. Consistent with this finding, a previous study of our group presents abnormal morphology of mitochondria in motor neurons of p40 wobbler mice [96]. Interestingly, in G93A SOD1 mice, a reduced protein expression of Nrf2 and GCLC in the cervical spinal cord has been reported [97]. Target genes of the Nrf2-ARE signaling pathway include *Gclc*, *Gclm*, *Gss*, *Ggt*, *Mrp1*, *Eaat3,* and *Slc7a11* [90,91]. Importantly, our results show tissue-specific dysregulations in most Nrf2 target genes. Reduced mRNA and protein levels of the antioxidant regulator Nrf2 have been reported in the postmortem primary motor cortex and spinal cord tissue of ALS patients [98]. For this reason, the Nrf2-ARE pathway has been estimated as a promising therapeutic target [99]. Numerous Nrf2 activators are being tested in pre-clinical studies for the treatment of ALS [100]. Edaravone, approved for the treatment of ALS in the US and Japan, has also been reported to activate the Nrf2 signaling pathway [101]. Nevertheless, Edaravone has been shown to clinically benefit only a small subset of ALS patients [102,103]. In addition to this, different Nrf2 activators, Fingolimod and Dimethyl fumarate, showed no observable benefit on ALS functional rating scale (ALSFRS) in Phase II randomized controlled trials [104,105]. Moreover, clinical trials aiming to improve glutathione contents in ALS by supplementation of glutathione or Acetlycysteine have been negative so far [106,107]. Consistent with these previous approaches, our study suggests that glutathione homeostasis is disrupted in the ALS model of the wobbler mouse and, thus, may be a therapeutic target. However, our data cannot be explained by Nrf2 activation or inactivation alone; as for the cervical spinal cord, some Nrf2 target genes are upregulated, while others are downregulated or unaltered. Our data suggest that future studies are needed to differentiate defects in glutathione metabolism and synthesis in the different cell types, especially astrocytes and neurons, with the objective of finding cell-specific therapeutic targets.

Unexpectedly, in all analyzed tissues, protein levels of GSS were significantly upregulated, independent of mRNA levels of *Gss*. A proteome-wide quantitative survey of in vivo ubiquitylation sites mapped 11.054 putative endogenous ubiquitylation sites on more than 4000 human proteins [108]. Among them, they identified a ubiquitylation side on human GSS. Ubiquitinylated TDP-43 containing aggregates have been reported in the spinal cord of wobbler mice [109], and extensive oxidative stress impairs the proteasome leading to protein aggregation, as reviewed by Shang and Taylor [110]. Given these facts, we hypothesized that GSS protein levels might be elevated in the examined tissues due to ubiquitylation and impaired proteasomal degradation. However, our preliminary results from the investigation of free ubiquitin and polyubiquitinylated proteins via Western Blot in the liver, cervical spinal cord, hippocampal, and cerebellar tissue of p40 wild-type and wobbler mice do not support this hypothesis (Appendix A). Thus, there has to be a different cause of the significant upregulation of GSS protein expression in all four tissues examined. Nefedova et al. reported that all-*trans* retinoic acid (ATRA) selectively induced expression of GSS via extracellular signal-regulated kinase 1/2 (ERK1/2) in myeloid-derived suppressor cells (MDSC) [111]. As GSS promoter activity was not affected in this process, post-transcriptional regulation of GSS is most likely [93,111]. However, the exact mechanism of ERK 1/2 on GSS expression and if this applies to the observed elevated protein levels in the respective tissues in wobbler mice remains to be examined.

## 5. Conclusions

Our results are in line with findings of oxidative stress and dysregulated glutathione levels in other ALS animal models [24,44,45] and ALS patients [1,42,43]. Our study indicates that the reduced glutathione amount obtained in the cervical spinal cord of wobbler mice is due to a decreased glutathione synthesis due to a decreased expression of the rate-limiting enzyme GCLC. For the first time, we provide evidence for a defective glutathione metabolism in the ALS in wobbler mice, which is not limited to motoric areas of the CNS. The detected decreased levels of glutathione in erythrocytes and plasma, besides alterations in glutathione metabolism and synthesis in all analyzed tissues of symptomatic wobbler mice, point toward the systemic character of ALS, as suggested before [112]. Although accumulating evidence suggests a direct or indirect involvement of aberrant glutathione homeostasis and metabolism in ALS [40], it remains unclear whether reduced glutathione is a causative factor or a result of neurodegeneration in ALS. So far, clinical trials aiming to improve glutathione contents in ALS by supplementation have been ineffective [106,107], as well as clinical trials targeting Nrf2 activating substances [102,103,104,105,113]. In consideration of our results, the dysregulations observed cannot be reversed by Nrf2 activation alone. Future studies are needed to differentiate defects in glutathione metabolism and synthesis in different cell types, especially astrocytes and neurons, in ALS. We conclude that the specific enhancement of glutathione synthesis might be a promising therapeutic target for ALS.

## Figures and Tables

**Figure 1 antioxidants-12-01014-f001:**
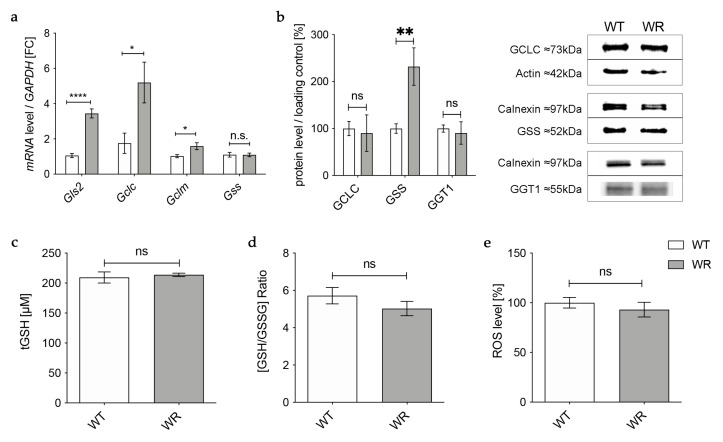
Physiological glutathione synthesis in liver of p40 wobbler mice. (**a**) mRNA expression levels of *Glutaminase 2* (*Gls2)*, *Gclc*, *Gclm*, and *Gss* from the distinct clinical phase (p40) of wild-type (WT) and wobbler (WR) liver were investigated by qPCR. Significantly increased mRNA levels of *Gls2*, *Gclc,* and *Gclm* were observed in the liver of WR, while mRNA expression levels of *Gss* were not significantly altered. For relative quantification, the 2^−∆∆Ct^ method was performed using *GAPDH* for normalization. FC = Fold Change; N = 4. (**b**) Left: Semiquantitative analysis of protein expression levels in the liver of p40 WT and WR mice. GSS protein expression is increased in liver of WR, while protein expression of GCLC and GGT1 remains unchanged. Right: Representative Western Blots of GCLC (73 kDa), GSS (52 kDa), and GGT1 (55 kDa) in the liver of p40 WT and WR mice. Actin (42 kDa) and Calnexin (97 kDa) were used as loading controls, respectively. Bar charts represent the semiquantitative analysis of protein expression levels; N = 6. (**c**,**d**) Levels of tGSH (**c**) as well as the ratio of GSH/GSSG (**d**) in liver of p40 WR mice show no significant alterations compared to WT. tGSH, GSH, and GSSG levels of p40 WR mice compared to WT mice were analyzed via a fluorometric glutathione assay; N = 4. (**e**) ROS levels in liver tissue of p40 WR mice are unchanged compared to age-matched WT controls. ROS levels in liver tissue of p40 WR and WT mice were analyzed via a fluorometric assay; N = 4. All data are provided as mean ± SEM. Data were tested for significance using Student‘s *t*-test. Significant differences are indicated by ns. *p* ≥ 0.05, * *p* < 0.05, ** *p* < 0.01, **** *p* < 0.0001.

**Figure 2 antioxidants-12-01014-f002:**
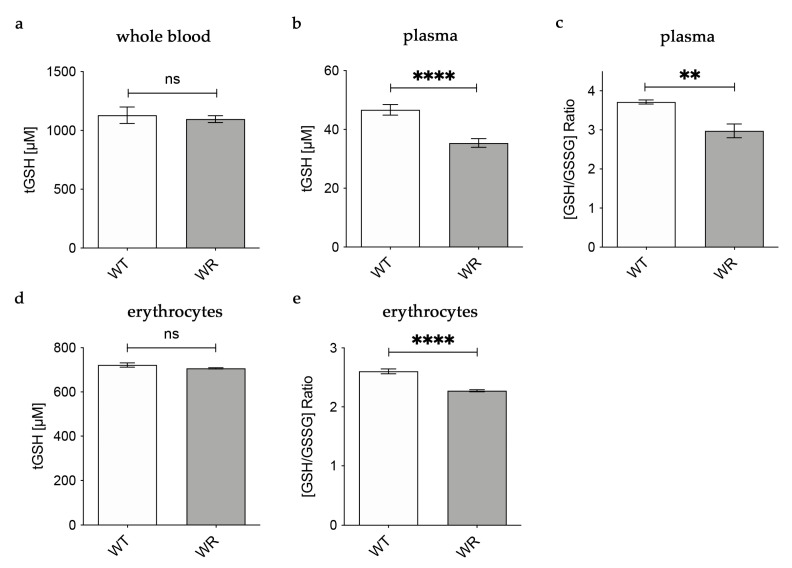
Decreased levels of glutathione in plasma of wobbler mice (p40). (**a**) HPLC measurement of tGSH displayed unaltered levels in whole blood of WR mice compared to WT mice; N = 6. (**b**) WR mice exhibit significantly lower levels of tGSH in plasma compared to WT mice; N = 7. (**c**) The ratio of GSH/GSSG is significantly decreased in plasma of WR mice compared to WT; N = 4. (**d**) Unaltered levels of tGSH in erythrocytes of p40 WR mice compared to WT mice; N = 4. (**e**) The ratio of GSH/GSSG is significantly decreased in erythrocytes of WR compared to WT; N = 4. All data are provided as mean ± SEM. Data were tested for significance using Student‘s *t*-test. Significant differences are indicated by ns. *p* ≥ 0.05, ** *p* < 0.01, **** *p* < 0.0001.

**Figure 3 antioxidants-12-01014-f003:**
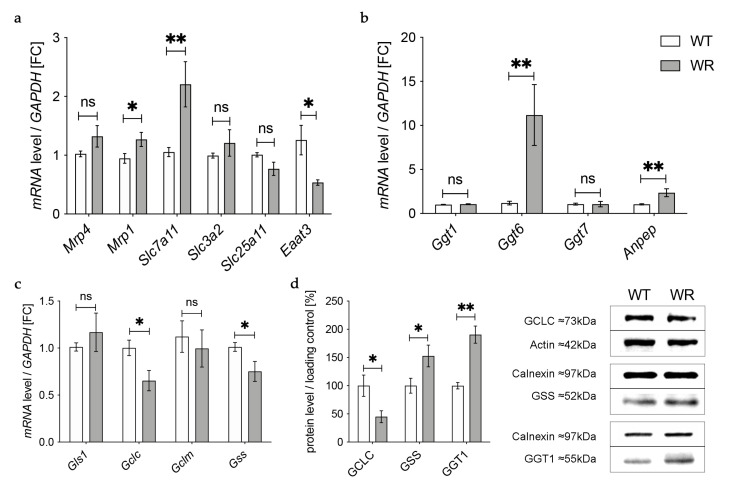
Altered glutathione metabolism and synthesis in the cervical spinal cord of wobbler mice. (**a**) mRNA expression levels of *Mrp4*, *Mrp1*, *Eaat3*, *Slc7a11*, *Slc3a2,* and *Slc25a11* in the cervical spinal cord of WT and WR mice from the distinct clinical stage (p40) were investigated by qPCR. mRNA levels of *Mrp1* and *Slc7a11* are significantly increased, while mRNA expression levels of *Eaat3* are significantly decreased. Unaltered mRNA levels of *Mrp4*, *Slc3a2*, and *Slc25a11*; N = 4–8. (**b**) Upregulation of genes involved in glutathione cleavage in the cervical spinal cord of wobbler mice (p40). Significantly increased mRNA levels of *Anpep* and *Ggt6* were measured, while mRNA expression levels of *Ggt1* and *Ggt7* are unaltered; N = 4. (**c**) Decreased mRNA expression levels of *Gclc* and *Gss*, alongside unaltered mRNA expression levels of *Gls1* and *Gclm* in the cervical spinal cord of WR mice compared to WT mice age p40. For relative quantification, the 2^−∆∆Ct^ method was performed using *GAPDH* for normalization. FC = Fold Change; N = 4. (**d**) Left: Semiquantitative analysis of protein expression levels in the spinal cord of p40 WT and WR mice. Right: Representative Western Blots of GCLC (73 kDa),GSS (52 kDa), and GGT1 (55 kDa) in the cervical spinal cord of p40 WT and WR mice. Actin (42 kDa) and Calnexin (97 kDa) were used as loading controls, respectively; N = 4–8. All data are provided as mean ± SEM. Data were tested for significance using Student‘s *t*-test. Significant differences are indicated by ns. *p* ≥ 0.05, * *p* < 0.05, ** *p* < 0.01.

**Figure 4 antioxidants-12-01014-f004:**
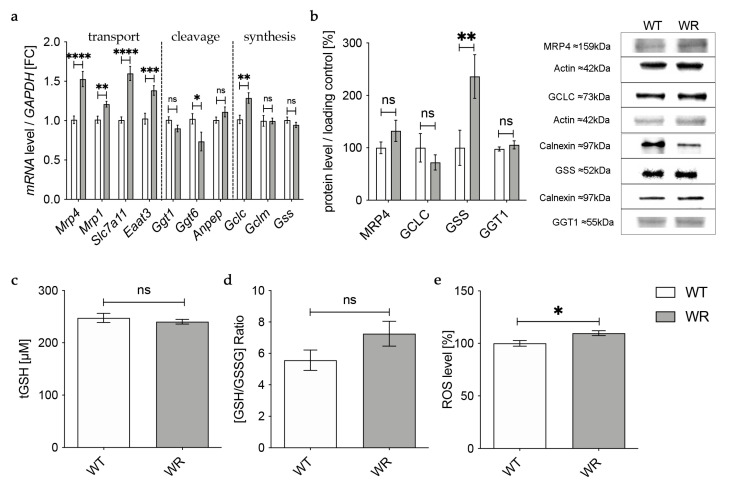
Physiological glutathione metabolism in hippocampus of wobbler mice. (**a**) mRNA expression levels of GSH transport relevant genes, *Mrp4*, *Mrp1*, *Slc7a11*, *Eaat3*, GSH cleavage relevant genes, *Ggt1*, *Ggt6*, *Anpep*, and GSH synthesis relevant genes, *Gclc*, *Gclm*, *Gss*, in the hippocampus of WT and WR mice from the distinct clinical stage (p40) were investigated by qPCR. Significantly increased mRNA levels of GSH transport-relevant genes were measured, while mRNA expression levels of *Ggt6* were significantly decreased. For relative quantification, the 2^−∆∆Ct^ method was performed using *GAPDH* for normalization; N = 4. (**b**) Left: Semiquantitative analysis of protein expression levels in the hippocampus of p40 WT and WR mice. Right: Representative Western Blots of MRP4 (159 kDa), GCLC (73 kDa), GSS (52 kDa), and GGT1 (55 kDa) in the hippocampus of p40 WT and WR mice. Actin (42 kDa) and Calnexin (97 kDa) were used as loading controls; N = 4. (**c**,**d**) tGSH (**c**) as well as [GSH/GSSG]-ratio (**d**) in hippocampus of p40 WR mice compared to WT mice was analyzed via a fluorometric glutathione assay. Neither tGSH nor the ratio of GSH/GSSG in hippocampus of WR mice is significantly altered compared to WT mice, age p40. N = 4. (**e**) ROS levels in hippocampus of WR and WT mice were detected via a fluorometric assay at the developmental stage p40. WR hippocampus shows significantly elevated levels of ROS compared to WT; N = 4. All data are provided as mean ± SEM. Data were tested for significance using Student‘s *t*-test. Significant differences are indicated by ns. *p* ≥ 0.05, * *p* < 0.05, ** *p* < 0.01, *** *p* < 0.001, **** *p* < 0.0001.

**Figure 5 antioxidants-12-01014-f005:**
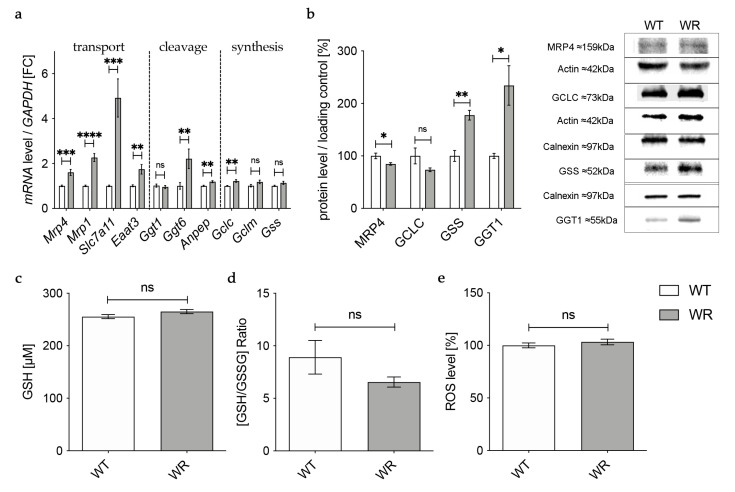
Physiological glutathione metabolism in cerebellum of WR mice. (**a**) mRNA expression levels of GSH transport relevant genes, *Mrp4*, *Mrp1*, *Slc7a11*, *Eaat3*, GSH cleavage relevant genes, *Ggt1*, *Ggt6*, *Anpep*, and GSH synthesis relevant genes, *Gclc*, *Gclm*, *Gss*, in the cerebellum of WT and WR mice from the distinct clinical stage (p40), were investigated by qPCR. Significantly increased mRNA levels of GSH transport, some cleavage relevant genes, as well as *Gclc* but no other GSH synthesis relevant genes. For relative quantification, the 2^−∆∆Ct^ method was performed using *GAPDH* for normalization; N = 4. (**b**) Left: Semiquantitative analysis of protein expression levels in the cerebellum of p40 WT and WR mice. Right: Representative Western Blots of MRP4 (159 kDa), GCLC (73 kDa), GSS (52 kDa), and GGT1 (55 kDa) in the cerebellum of p40 WT and WR mice. Actin (42 kDa) and Calnexin (97 kDa) were used as loading controls; N = 4. (**c**,**d**) tGSH (**c**) as well as [GSH/GSSG]-ratio (**d**) in cerebellum of p40 WR mice compared to WT mice was analyzed via a fluorometric glutathione assay. Neither tGSH nor the ratio of GSH/GSSG in cerebellum of WR mice is significantly altered compared to WT mice at the age of p40; N = 4. (**e**) Unaltered levels of reactive oxygen species in cerebellum of wobbler mice compared to wild-type were detected via a fluorometric assay at the developmental stage p40; N = 4. All data are provided as mean ± SEM. Data were tested for significance using Student‘s *t*-test. Significant differences are indicated by ns. *p* ≥ 0.05, * *p* < 0.05, ** *p* < 0.01, *** *p* < 0.001, **** *p* < 0.0001.

**Figure 6 antioxidants-12-01014-f006:**
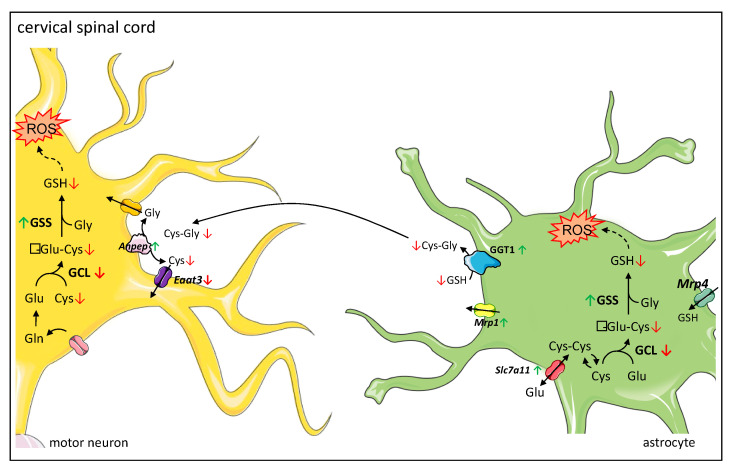
Proposed mechanism of dysregulated glutathione metabolism and synthesis in cervical spinal cord of wobbler mice. Glutathione metabolism between astrocytes and neurons: astrocytes import glutathione (GSH) via MRP4 (Multidrug Resistance Protein 4), while neurons do not contain an uptake mechanism for GSH and therefore rely on their own GSH synthesis. Both astrocytes and neurons can synthesize GSH from its constituent amino acids. Astrocytes can import cystine via the cystine/glutamate exchanger system Xc-. Astrocytes supply neurons with substrates for GSH synthesis. They export GSH via MRP1 (Multidrug Resistance Protein 1). In the next step, the ectoenzyme GGT (Gamma-Glutamyl Transferase) hydrolyzes GSH. The resulting Cysteinyl–Glycine (Cys-Gly) is cleaved by the ectoenzyme Aminopeptidase N (ANPEP). Next, the resulting amino acids are imported into neurons by different transporters, where they can be used for GSH synthesis. The main transporter for cysteine uptake in neurons is EAAT3 (Excitatory Amino Acid Transporter 3). Proposed mechanism of dysregulated glutathione metabolism and synthesis in the cervical spinal cord: a decreased protein expression of GCL (γ-Glutamyl-cysteinyl-ligase) leads to a decreased synthesis of γ-Glu-Cys (γ-Glutamyl-Cystein), resulting in a decreased GSH synthesis, even though protein expression of GSS (glutathione synthetase) is increased. Thus, less GSH is available for export from astrocytes via MRP1 and cleavage by GGT and ANPEP. As a compensatory mechanism, GGT1 is upregulated. However, EAAT3 is downregulated, which aggravates the deficient cysteine supply to neurons. Overall, this leads to an insufficient GSH synthesis in neurons. Due to the inadequate GSH levels, neurons’ and astrocytes’ defense against ROS (reactive oxygen species) is severely impaired. In astrocytes, increased levels of ROS lead to an increased expression of *Slc7a11* encoding for the light chain of system Xc-, enhancing cystine import in astrocytes. Legend: Green arrows indicate an upregulation; red arrows indicate a downregulation of corresponding enzyme on protein or mRNA levels. Cys-Cys: Cystine; Glu: Glutamate; Gln: Glutamine.

**Table 1 antioxidants-12-01014-t001:** Specific primers used for quantitative PCR.

Gene (Gene ID)	Sense Primer	Antisense Primer
*Anpep (16790)*	5′-TGGAGGATCTTTCTCCTTTGCC-3′	5′-GTGGCTGAGTTATCCGCTTT-3′
*Eaat3 (20510)*	5′-TTCCTACGGAATCACTGGCTG-3′	5′-TGTGTCCTCGAACCACGACT-3′
*Gapdh (14433)*	5′-GGAGAAACCTGCCAAGTATGA-3′	5′-TCCTCAGTGTAGCCCAAGA-3′
*Gclc (14629)*	5′-ACAAGGACGTGCTCAAGTGG-3′	5′-GTCTCAAGAACATCGCCTCCA-3′
*Gclm (14630)*	5′-CCCCGATTTAGTCAGGGAGTTT-3′	5′-TTTCATCGGGATTTATCTTCTCCAC-3′
*Ggt1 (14598)*	5′-GAAGCCCGACCACGTGTACT-3′	5′-CCACGGAACCACCTTCCTGT-3′
*Ggt6 (71522)*	5′-ACAAGCTACAACTCTGGGAGC-3′	5′-CTCCTTAGGGAGAGGACCAG-3′
*Ggt7 (207182)*	5′-TTCGTCGTCATCGGAGATGG-3′	5′-GCAGCTGAGAATGGGTCCTT-3′
*Gls1 (14660)*	5′-GGTCTTCCTGCAAAATCTGGA-3′	5′-CCTTAACACTGTTGCCCATCT-3′
*Gls2 (216456)*	5′-ACAAGACCGTGGTGAACCTG-3′	5′-GGCTGTGCGGGAATCATAGT-3′
*Gss (14854)*	5′-AGGACGACTATACTGCCCGT-3′	5′-AATCTGAGCGATTCAGGCCC-3′
*Mrp1 (17250)*	5′-GCTGGGCAGACCTCTTCTAC-3′	5′-CAGTGTTGGGCTGACCAGTA-3′
*Mrp4 (239273)*	5′-CGTTTGCTGACCTCATTGCC-3′	5′-ACGCCATGTTCATCCCTCTG-3′
*Slc3a2 (17254)*	5′-TGGTTATCATCGTTCGGGCG-3′	5′-GCCTACAAAGGCCTGAAGGT-3′
*Slc7a11 (26570)*	5′-CACTTTTTGGAGCCCTGTCC-3′	5′-CCAGCAAAGGACCAAAGACC-3′
*Slc25a11 (67863)*	5′-CTGGTGCATTTGTGGGAACG-3′	5′-ACATTTTTGTAGCCACGGCG-3′

**Table 2 antioxidants-12-01014-t002:** Primary and secondary antibodies used for Western Blotting.

Antibody	Order Number	Dilution	Membrane
Anit-γGCSc mouse monoclonal IgG_2a_ antibody	#sc-390811, Santa Cruz, USA	1:200 in ROTI^®^-Block (#A151, Roth, Karlsruhe, Germany)	Nitrocellulose membrane (#1704270, biorad)
Anti-GSS mouse monoclonal IgG_1_ antibody	#sc-166882, Santa Cruz, USA	1:200 in ROTI^®^-Block (#A151, Roth, Karlsruhe, Germany)	Nitrocellulose membrane (#1704270, biorad)
Anti-MRP4 rabbit monoclonal IgG antibody	#12705, Cell Signaling Technology, USA	1:1000 in 5% BSA (#8076.2, Roth) in TBS-T	PVDF membrane (#1704272, biorad)
Anti-GGT1 mouse monoclonal IgG_2b_ antibody	#sc-374495, Santa Cruz, USA	1:1000 in 5% BSA (#8076.2, Roth) in TBS-T	Nitrocellulose membrane (#1704270, biorad)
Anti-Actin rabbit polyclonal IgG antibody	#A5060, Sigma, St. Louis, MO, USA	1:250 in ROTI^®^-Block (#A151, Roth, Karlsruhe, Germany)	Nitrocellulose membrane (#1704270, biorad) or PVDF membrane (#1704272, biorad)
Anti-Calnexin rabbit polyclonal IgG antibody	#sc-11397, Santa Cruz, USA	1:200 in ROTI^®^-Block (#A151, Roth, Karlsruhe, Germany)	Nitrocellulose membrane (#1704270, biorad)
Horse anti-mouse IgG (H + L)-Horseradish Peroxidase antibody	#PI-2000, Vector Laboratories, Bulingame, CA, USA	1:10,000 in TBS-T	
Goat anti-rabbit IgG (H + L)-Horseradish Peroxidase conjugate	#1706515, BioRad, Hercules, CA, USA	1:10,000 in TBS-T	

## Data Availability

All data generated in this study are published in this article.

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
