# Peer review of "Defects in Glutathione System in an Animal Model of Amyotrophic Lateral Sclerosis"

_antioxidants, 2023, doi:10.3390/antiox12051014_

Round 1

Reviewer 1 Report

In this manuscript, entitled “Defects in Glutathione-System in an Animal Model of Amyotrophic Lateral Sclerosis”, Wunsch et al. aimed to dissect the causal roles of decreased glutathione levels in the CNS using the ALS model wobbler mouse. The authors show the decreased of the expression level of enzymes involved in glutathione synthesis in spinal cord, hippocampus, cerebellum, liver, and blood samples of the ALS model, using qPCR, Western Blot, HPLC and fluorometric assays, and interpreted that the systematic deficient is the reason for inefficient antioxidative system leads to ROS elevation. 

There are quite a lot of single cell sequencing studies focused on ALS and its animal models, which taking away the novelty of this study a lot. The natural of heterogeneity of tissues make the results from this study not very helpful.  

Reviewer 2 Report

This manuscript is an extension of previous work from the same lab on disruption of glutathione (GSH) homeostasis in the spinal cord of wobbler mice, a model of ALS. In the current study, the authors assess the levels of GSH and gene transcripts for a number of different proteins involved in GSH synthesis and metabolism in multiple tissues to determine whether or not there are systemic changes in GSH metabolism in these mice and to identify the enzymes and pathways involved. Thus, this manuscript is of significant interest since there are still no effective treatments for ALS so understanding how GSH metabolism is altered in a disease model could lead to new therapeutic approaches. The study is quite thorough in that the authors examine multiple tissues, including liver, spinal cord, hippocampus and cerebellum, as well as the expression of enzymes directly and indirectly involved in GSH synthesis and extracellular breakdown as well as transporters involved in the import of substrates or GSH itself. However, the manuscript has one significant problem and that is for most of the proteins the authors only measure the gene transcripts that encode these proteins, not the levels of the proteins themselves. This would not be problematic if the authors were able to show that the gene transcripts and protein levels trend in the same directions for those proteins where both were assessed. However, this is not the case. In all of the tissues examined, when the authors compared the levels of the gene transcript for GSS between wt and wobbler mice, they found either no difference or a significant decrease. However, the levels of GSS protein were increased in all of the same tissues. A similar dichotomy was seen with MRP4 gene and protein expression in the hippocampus and GCLC gene and protein expression in the liver, hippocampus and cerebellum. This dichotomy between gene and protein expression means that the authors cannot equate conclusions based on gene expression to those based on protein expression. In other words, the authors need to do Western blots for all of the proteins examined in all of the tissues tested to be able to draw accurate conclusions regarding how GSH metabolism is altered in the ALS model. There are a several other points that also need to be addressed as listed below. Finally, the manuscript needs thorough editing as there are numerous word use errors scattered throughout.

1. Why were different proteins used to normalize the Western blots for GCLC and GSS? The authors need to provide a rationale for this choice.

2. Why were two different methods used to measure total GSH? In addition, the description of the use of the GSH/GSSG detection kit is not clear. How can the same assay mix be used to measure both reduced GSH and total GSH? Also, the authors need to specify how GSSG was calculated.

3. Discussion, 1st paragraph, pg 15: The discussion about ROS in this paragraph needs to be consolidated rather than mentioned both at the beginning and end of the paragraph.

4. Discussion, 2nd paragraph, pg. 15: This description of the potential role of Nrf2 is much too long, especially as the authors correctly conclude that their results cannot be explained by Nrf2 activation or inactivation alone. This paragraph could be shortened considerably and better focused on this conclusion rather than all of the previously published data.

5. Discussion, 2nd paragraph, pg. 16: Why don’t the authors test whether GSS is more ubiquinated in the wobbler mouse? It should be straightforward to immunoprecipitated GSS and blot with anti-ubiquitin antibodies. Alternatively, as a first look, the authors could simply check to see if there are more ubiquinated proteins in the tissue extracts from the wobbler mice as compared to the wt mice.

Reviewer 3 Report

This is an interesting and important topic and a potentially interesting manuscript; the reference section is nicely done.  However, revisions and clarifications are needed in the materials and methods sections.  In addition, some grammar and word choice need changes.  I have indicated some of these issues below:

Line 89: add ‘the’  before the word ‘molecular’

Line 180: what centrifuge was used? (since ‘top speed’ does not allow one to reproduce your method).

Also in line 180, spell out 50 as fifty

Always spell out abbreviations first time used (see line 197 DCFH as an example)

line 217:  should this line end in  ‘..as shown in Figure 1’   ??  Figure 1 also shows other information which should be indicated in the text.   In figure 1, in some cases Gclc is shown and in other cases GCLC is shown.  The authors need to clearly indicate what they are trying to convey.  The 1b Western Blots insert is very confusing; are the authors indicating each band pair (for WTl and WR) are on the same gel as the bands below?   This must be clarified here and for other figures especially since the authors appear to be concluding quantitative differences which are very hard to tell from these figures.  In the legend from Figure 1 also clarify that 1d is not ROS (this may be 1e) since 1 d is the GSH/GSSG ratio.  Also what do authors mean by word ‘exemplary (line 226 and other places in other figures)…do they mean ‘representative’?   How do the authors define’ semiquantitative’ as in line 224? and other places in legends. Line 234 capitalize Students’ t-test.

Line 243: move to line 240 (following (p40) for clarity.

Line 246 do authors mean (d) rather than (c) here? Whereas in line 247 is 2 c indicated rather than 2d?  In line 247 what do authors mean by use of word ‘convenient? For figure 2 add d and e indicating which panels to look at; als0 line 254  is likely to be d and not c.  This figure legend should be corrected.

Line 296: what % of test genes were up-regulated (it does not appear to me to be ‘most’.

Line 305: how did authors quantitate the bands from the Western Blots (and thus conclude significantly increased protein expression)??? Since the bands (figure 3) look about the same in the data presented as bands.

Line320: what do the authors mean by ‘exemplary’ here and other places in the legends

Line 342:  add’ modest but…’ significant increase to this line

Line 372: is ROS in figure 5e and not 5d as indicated?  Add d and e to figure 5 also make legend correct

I recommend moving  lines 403-413 to the introduction section so that the reader has more context for the experiments being done and especially with the extensive use of abbreviations used by the authors. 

Line 419 and other places in the text:  et al. is more correct than et al

Lines 442 and 452 are redundant

Line 469 : the authors indicate ANpep is upregulated but they should indicate only in some areas

Line 507 spell out fifty percent rather than use numeral to start a sentence; same comment for line 508

Line 584: add ‘…is due to’ for clarity

Line 586:  what do the authors mean by word ‘Convenient’?

Line 624: do authors mean ‘sites’ or ‘sides’ in this sentence?

Line 628:  the authors are Shang and Taylor for this reference

Round 2

Reviewer 1 Report

My major concern with this manuscript is the novelty. The author convinced me that the direct comparison across tissues, and regions in one study is useful and could provide insights. I agree this manuscript could be accepted for publishing. 

Reviewer 2 Report

The authors have made an excellent effort to address the concerns raised in my original review. I think that the additions and changes that they have made significantly improve the manuscript and answer all of my questions.

Reviewer 3 Report

I thank the authors for their careful revisions and think the manuscript is much easier to read and understand.  Interesting work.